# AutoLoRA: AutoGuidance Meets Low-Rank Adaptation for Diffusion Models

## Abstract

Low-rank adaptation (LoRA) is a fine-tuning technique that can be applied to conditional generative diffusion models. LoRA utilizes a small number of context examples to adapt the model to a specific domain, character, style, or concept. However, due to the limited data utilized during training, the fine-tuned model performance is often characterized by strong context bias and a low degree of variability in the generated images. To solve this issue, we introduce AutoLoRA, a novel guidance technique for diffusion models fine-tuned with the LoRA approach. Inspired by other guidance techniques, AutoLoRA searches for a trade-off between consistency in the domain represented by LoRA weights and sample diversity from the base conditional diffusion model. Moreover, we show that incorporating classifier-free guidance for both LoRA fine-tuned and base models leads to generating samples with higher diversity and better quality. The experimental results for several fine-tuned LoRA domains show superiority over existing guidance techniques on selected metrics.

## 1 Introduction

The key aspects of image-generating diffusion models focus on image quality, the variability in the results, and the production of images tailored to specific conditions (Zhang et al., 2023a). Classifier-free guidance (CFG) (Ho & Salimans, 2022) is a sampling method for generating diverse, good-quality images utilizing a conditional diffusion model. The idea of this approach is to use a combination of constrained and unconstrained diffusion models to find the trade-off between diversity and consistency with the conditioning factor.

In AutoGuidance (Karras et al., 2024), the fully-trained diffusion model is conditioned using a less-trained version. This approach parallels the technique employed in Classifier-Free Guidance (CFG), where two models are combined during the sampling process. However, unlike CFG, which uses conditioned and unconditioned models, AutoGuidance leverages different stages of training within a single model. The fully trained model, which tends to overfit to the training data, is counterbalanced by the partially trained model, thereby introducing greater diversity in the generated outputs. This approach is critical for enhancing the variability of generated images, a fundamental requirement for improving generative model performance.

Low-Rank Adaptation (LoRA) (Hu et al., 2021) is a fine-tuning technique widely applied in large diffusion models to adjust them to some specific dataset and enforce generating images in particular style, character, or some other specific concepts. However, the adaptation process is usually made using a relatively small number of data examples, which causes a small degree of variability in the output, and the tuned model is biased towards the training examples.

In this paper, we present AutoLoRA, which allows for an increase in the variety of images generated from LoRA models and a reduction of data bias. AutoLoRA use the general idea of AutoGuidance, which claims that overfit model can be improved by conditioning by the model with a lower quality but greater diversity. AutoLoRA increase LoRA generation variability by conditioning LoRA with a base diffusion model. In practice, we use a model before LoRA tuning to condition the final variant fine-tuned with the LoRA approach. In this work, we focus on diffusion models with additional context inputs, like prompts. For such an approach, we additionally apply classifier-free guidance for both base and fine-tuned versions of the diffusion models. With this approach, we incorporate an additional level of regularisation and new degrees of freedom to increase the variety of the details

in the image that are not controlled by context inputs. As a consequence, we observe an increase in the prompt alignment and diversity of generated samples.

The following constitutes a list of our key contributions. First, we show that the CFG and AutoGuidance mechanism can solve a problem with a low variety of generated images in the LoRA model. Second, we provide a guidance approach that enables us to find the trade-off between prompt adjustment, LoRA consistency, and generalization. Third, AutoLoRA reduces bias in the LoRA model caused by using a relatively small data set to tune the model.

## 2 RELATED WORK

In this section, we review the related work, starting with the development of general diffusion models, followed by an overview of guided diffusion models, and concluding with methods for low-rank adaptations.

**Diffusion models**   The concept of diffusion models in the deep learning community was first introduced in Sohl-Dickstein et al. (2015). Utilizing Stochastic Differential Equations (SDEs), diffusion models enable the transformation of a simple initial distribution (e.g., a normal distribution) into a more complex target distribution through a series of tractable diffusion steps. Subsequent advancements, such as reducing the number of trajectory steps (Bordes et al., 2017), led to more efficient diffusion models.

A major breakthrough occurred with the introduction of Denoising Diffusion Probabilistic Models (DDPMs) in (Ho et al., 2020; Dhariwal & Nichol, 2021). DDPMs employ a weighted variational bound objective by combining diffusion probabilistic models with denoising score matching (Song & Ermon, 2019). While these models exhibited strong generative performance (i.e., high-quality samples), their computational cost remained a significant limitation.

The first notable improvement in terms of scalability, particularly sample efficiency, came from generalizing DDPMs to non-Markovian diffusion processes, resulting in shorter generative Markov chains, called Denoising Diffusion Implicit Models (DDIMs) (Song et al., 2020).

Finally, the high computational cost of scaling diffusion models to high-dimensional problems was alleviated by Latent Diffusion Models (Rombach et al., 2022), which proposed performing diffusion in the lower-dimensional latent space of an autoencoder. One notable example of a latent diffusion model is Stable Diffusion (Rombach et al., 2022), which demonstrated the practical application of this approach. Further improvements in scalability were achieved in models like SDXL (Podell et al., 2023), which extended the capabilities of latent diffusion models to even larger and more complex tasks.

**Guidance of diffusion models**   In diffusion models, the underlying Stochastic Differential Equation (SDE) process plays a crucial role. While it enables strong generative performance, improved scalability, and faster training compared to models based on Ordinary Differential Equations (ODEs) (Dinh et al., 2014; Rezende & Mohamed, 2015; Grathwohl et al., 2018), the stochastic inference process requires *guidance* to produce desirable samples. To steer the generation process in a preferred direction, several guidance techniques have been developed, which can be broadly categorized based on different strategies: classifier-based guidance, Langevin dynamics, Markov Chain Monte Carlo (MCMC), external guiding signals, architecture-specific features, and others. Despite their differences, most of these techniques direct the diffusion process toward regions of minimal energy, estimated through various proxies.

One of the most well-known techniques is Classifier Guidance (Dhariwal & Nichol, 2021; Poleski et al., 2024), which uses an external classifier trained to predict the class from noisy intermediate diffusion steps. In contrast, Classifier-Free Guidance (Ho & Salimans, 2022) eliminates the need for a separate classifier by training a single model in two modes—conditioned and unconditioned.

Other approaches are inspired by sampling techniques. For instance, in the diffusion sampling community, Langevin dynamics is commonly used for off-policy steering, where at each trajectory step, the model follows the scaled gradient norm toward regions of lowest energy (the highest *log probability*), as demonstrated in works like Zhang & Chen (2021) and Sendera et al. (2024). Alternatively,

MCMC-based sampling strategies are also applied directly in diffusion processes (Song et al., 2023; Chung et al., 2023).

A few methods incorporate external guiding functions to smooth the generative trajectory toward desired outcomes, such as in Bansal et al. (2023). Others leverage architectural properties of diffusion models, for example, using intermediate self-attention maps (Hong et al., 2023) or training an external discriminator network (Kim et al., 2022).

Most recently, AutoGuidance (Karras et al., 2024) extends classifier-free guidance by replacing the unconditional model with a smaller, less-trained version of the model itself to guide the conditional one. Our approach is inspired by AutoGuidance but diverges by introducing small adapters (i.e., LoRA modules) to enhance the diversity of generated samples in diffusion models.

**Parameter-efficient fine-tuning and LoRA**   With the increasing size of modern deep learning models, full fine-tuning for downstream tasks is becoming increasingly impractical, and this challenge will only grow as models scale further. This has led to the rise of Parameter-Efficient Fine-Tuning (PEFT) methods, which typically involve adding small, trainable modules to a pretrained model, such as Adapters (Houlsby et al., 2019).

Among PEFT techniques, Low-Rank Adaptation (LoRA) (Hu et al., 2021) has emerged as the most widely adopted solution, originating from the large language model (LLM) community. LoRA introduces low-rank adaptations to each weight matrix, factorizing updates into two low-rank matrices. This allows LoRA to achieve fine-tuning results comparable to full fine-tuning, but with significantly fewer parameters (ranging from 100 to even 10,000 times smaller).

Due to its effectiveness, numerous LoRA variants have been proposed. For example, methods like Dy-LoRA (Valipour et al., 2022), AdaLoRA (Zhang et al., 2023c), and IncreLoRA (Zhang et al., 2023b) dynamically adjust the rank hyperparameter. GLoRA (Chavan et al., 2023) generalizes LoRA by incorporating a prompt module, while Delta-LoRA (Zi et al., 2023) simultaneously updates pretrained model weights using the difference between LoRA weights. Further model size reductions have been achieved with QLoRA (Dettmers et al., 2023). Additionally, ongoing research continues to propose new LoRA-based approaches (including, e.g., Kopiczko et al. (2023); Hao et al. (2024); Hayou et al. (2024); Zhang & Pilanci (2024)) and deepen theoretical understanding (Fu et al., 2023; Jang et al., 2024).

Most importantly for our work, LoRA has proven to be highly versatile and is now commonly applied in diffusion models (Ryu, 2023; Smith et al., 2023; Choi et al., 2023; Zheng et al., 2024). Given the variety of possible LoRA-based methods, we follow the widely adopted and well-understood procedure outlined in the original LoRA paper by Hu et al. (2021).

## 3   PRELIMINARIES

In this section, we present our AutoLoRA method dedicated for AutoGuidance of LoRa models. We start by introducing LoRa, and then show how the Classifier-free guidance (CFG) and AutoGuidance work. At the end we show how to utilize AutoLoRA to increase variability in LoRa models.

**Diffusion Models**   Diffusion models, such as Denoising Diffusion Probabilistic Models (DDPMs) (Ho et al., 2020; Dhariwal & Nichol, 2021), operate by modeling a Markov chain of successive noise addition and denoising steps. These models involve a forward process, where noise is gradually added to data, and a reverse process, where the model learns to remove noise, generating high-quality samples from a noise vector.

The forward process is defined as a series of Gaussian noise steps applied to a data sample $\mathbf{x}_0$, transitioning it into increasingly noisy versions $\mathbf{x}_t$ as $t$ progresses from $0$ to $T$. This process can be described as:

$$q(\mathbf{x}_t|\mathbf{x}_{t-1}) = \mathcal{N}(\mathbf{x}_t; \sqrt{1-\beta_t}\mathbf{x}_{t-1}, \beta_t\mathbf{I}) \tag{1}$$

where $\{\beta_t \in (0,1)\}_{t=1}^T$ is a noise schedule parameter that controls the level of noise added at step $t$.

The reverse process, modeled by the diffusion model, attempts to reconstruct $\mathbf{x}_0$ from a noisy $\mathbf{x}_T$ by progressively denoising it. The goal of training is to learn a model $p_\theta(\mathbf{x}_{t-1}|\mathbf{x}_t)$ that can reverse the

---

**Algorithm 1** Reverse Diffusion with CFG

---

**Require:** $\mathbf{x}_T \sim \mathcal{N}(0, \mathbf{I}_d), 0 \leq \omega \in \mathbb{R}$, conditioning factor $y$

**for** $t = T$ to 1 **do**

$\quad \hat{\boldsymbol{\epsilon}}^w(\mathbf{x}_t, y) = \hat{\boldsymbol{\epsilon}}(\mathbf{x}_t) + w \cdot (\hat{\boldsymbol{\epsilon}}(\mathbf{x}_t, y) - \hat{\boldsymbol{\epsilon}}(\mathbf{x}_t, \emptyset))$

$\quad \hat{\mathbf{x}}^w(\mathbf{x}_t, y) = (\hat{\mathbf{x}}_t - \sqrt{1 - \bar{\alpha}_t} \cdot \hat{\boldsymbol{\epsilon}}^w(\mathbf{x}_t, y)) / \sqrt{\alpha_t}$

$\quad \mathbf{x}_{t-1} = \sqrt{\bar{\alpha}_{t-1}} \cdot \hat{\mathbf{x}}^w(\mathbf{x}_t, y) + \sqrt{1 - \bar{\alpha}_{t-1}} \cdot \hat{\boldsymbol{\epsilon}}^w(\mathbf{x}_t, y)$

**end for**

**return** $x_0$

---

**Algorithm 2** Reverse Diffusion with AutoLoRA

---

**Require:** $\mathbf{x}_T \sim \mathcal{N}(0, \mathbf{I}_d), 0 \leq w_1, w_2, \gamma \in \mathbb{R}$, conditioning factor $y$

**for** $t = T$ to 1 **do**

$\quad \hat{\boldsymbol{\epsilon}}^{w_1}(\mathbf{x}_t, y) = \boldsymbol{\epsilon}(\mathbf{x}_t, \emptyset) + w_1 \cdot (\boldsymbol{\epsilon}(\mathbf{x}_t, y) - \boldsymbol{\epsilon}(\mathbf{x}_t, \emptyset))$

$\quad \hat{\boldsymbol{\epsilon}}_{\text{LoRA}}^{w_2}(\mathbf{x}_t, y) = \boldsymbol{\epsilon}_{\text{LoRA}}(\mathbf{x}_t, \emptyset) + w_2 \cdot (\boldsymbol{\epsilon}_{\text{LoRA}}(\mathbf{x}_t, y) - \boldsymbol{\epsilon}_{\text{LoRA}}(\mathbf{x}_t, \emptyset))$

$\quad \hat{\boldsymbol{\epsilon}}_{\text{AutoLoRA}}^{\gamma, w_1, w_2}(\mathbf{x}_t, y) = \hat{\boldsymbol{\epsilon}}^{w_1}(\mathbf{x}_t, y) + \gamma \cdot (\hat{\boldsymbol{\epsilon}}_{\text{LoRA}}^{w_2}(\mathbf{x}_t, y) - \hat{\boldsymbol{\epsilon}}^{w_1}(\mathbf{x}_t, y))$

$\quad \hat{\mathbf{x}}^\gamma(\mathbf{x}_t, y) = (\hat{\mathbf{x}}_t - \sqrt{1 - \bar{\alpha}_t} \cdot \hat{\boldsymbol{\epsilon}}_{\text{AutoLoRA}}^\gamma(\mathbf{x}_t, y)) / \sqrt{\alpha_t}$

$\quad \mathbf{x}_{t-1} = \sqrt{\bar{\alpha}_{t-1}} \cdot \hat{\mathbf{x}}^\gamma(\mathbf{x}_t, y) + \sqrt{1 - \bar{\alpha}_{t-1}} \cdot \hat{\boldsymbol{\epsilon}}_{\text{AutoLoRA}}(\mathbf{x}_t, c)$

**end for**

**return** $x_0$

---

noise process. In practice, the model is often parametrized as $\epsilon_\theta(\mathbf{x}_t, t)$ and is trained to predict real noise applied on $\mathbf{x}_0$ following Equation 1.

In conditional generation, the model is conditioned on additional information such as a label or text prompt, denoted as $y$. The model learns a conditional distribution $p_\theta(\mathbf{x}_{t-1} | \mathbf{x}_t, y)$, which incorporates the conditioning information into the generative process.

**Low-Rank Adaptation (LoRA)**   Low-Rank Adaptation (LoRA) (Hu et al., 2021) is a parameter-efficient fine-tuning technique designed to adapt pre-trained models to new tasks by introducing low-rank modifications to their weight matrices. LoRA was initially proposed to address the challenge of fine-tuning large language models (LLMs) while significantly reducing the computational and memory overhead associated with updating full sets of model parameters.

The application of LoRA to diffusion models is particularly advantageous in scenarios where large pre-trained models are adapted to specific tasks or datasets with limited computational resources. By leveraging LoRA, it becomes feasible to adapt diffusion models for new tasks (e.g., domain-specific image generation or text-guided diffusion) without the need for full-scale retraining.

LoRA leverages the observation that the learned weight matrices in large-scale models, particularly in attention-based architectures, often reside in a lower-dimensional subspace. Instead of updating the full model weights, LoRA freezes the pre-trained parameters and introduces low-rank matrices that are added to the weight updates during training. Specifically, LoRA decomposes the weight update matrices into two smaller matrices, effectively reducing the number of parameters that need to be trained and stored.

Mathematically, a weight matrix $W \in \mathbb{R}^{d \times k}$, where $d$ is the input dimension and $k$ is the output dimension, is modified by adding a low-rank matrix update as follows:

$$W' = W + \alpha \cdot \Delta W = W + \alpha \cdot A \cdot B \tag{2}$$

where $A \in \mathbb{R}^{d \times r}$ and $B \in \mathbb{R}^{r \times k}$, where $r$ is the rank, typically chosen such that $r \ll d, k$, resulting in a significant reduction in the number of parameters that need to be optimized. This low-rank adaptation allows for efficient fine-tuning while maintaining most of the representational power of the original model. The $\alpha$ represents the LoRA scaling parameter that controls the impact of LoRA fine-tuning weights, mainly during inference.

**Classifier-Free Guidance** Classifier-Free Guidance (CFG) (Ho & Salimans, 2022) is a technique used in diffusion models to steer the generative process more effectively without relying on external classifiers. It has proven highly effective in improving the quality of generated samples in various tasks such as image and text generation.

Before the introduction of Classifier-Free Guidance, diffusion models often employed classifier-based guidance (Dhariwal & Nichol, 2021). In this setup, an external classifier $c_\phi(y|\mathbf{x}_t)$, trained to predict the conditioning label $y$ from intermediate noisy samples $\mathbf{x}_t$, was used to steer the reverse process. This guidance was achieved by modifying the reverse sampling step as:

$$\hat{\boldsymbol{\epsilon}}_\theta(\mathbf{x}_t) = \boldsymbol{\epsilon}_\theta(x_t) - \sqrt{1 - \bar{\alpha}_t}\, w \nabla_{\mathbf{x}_t} \log c_\phi(y|\mathbf{x}_t) \tag{3}$$

where $w$ is a scaling factor that adjusts the strength of the classifier's influence, and $\bar{\alpha}_t = \prod_{i=1}^{t} 1 - \beta_i$. While effective, classifier-based guidance introduces several downsides, such as added complexity and potential inaccuracies due to classifier errors.

Classifier-Free Guidance offers a simpler and more robust alternative by eliminating the need for an external classifier. Instead, the diffusion model itself is trained in two modes – *conditional* and *unconditional*:

- *conditional mode* - the model is trained to predict the denoised data $\mathbf{x}_0$ given noisy data $\mathbf{x}_t$ and conditioning information $y$, learning the conditional distribution $p_\theta(\mathbf{x}_{t-1}|\mathbf{x}_t, y)$;

- *unconditional mode* - the same model is also trained without any conditioning, learning the unconditional distribution $p_\theta(\mathbf{x}_{t-1}|\mathbf{x}_t)$.

During inference, CFG uses a combination of the conditional and unconditional predictions to guide the generation (see Algorithm 1). Specifically, for a given noisy sample $\mathbf{x}_t$, the guidance is achieved by interpolating between the conditional and unconditional predictions as follows:

$$\hat{\boldsymbol{\epsilon}}_\theta^w(\mathbf{x}_t, y) = \boldsymbol{\epsilon}_\theta(\mathbf{x}_t, \emptyset) + w\left(\boldsymbol{\epsilon}_\theta(\mathbf{x}_t, y) - \boldsymbol{\epsilon}_\theta(\mathbf{x}_t, \emptyset)\right), \tag{4}$$

where $\boldsymbol{\epsilon}_\theta(\mathbf{x}_t, \emptyset)$ is the model's prediction of the noise in $\mathbf{x}_t$ when no conditioning is provided (unconditional), $\boldsymbol{\epsilon}_\theta(\mathbf{x}_t, y)$ is the prediction of the noise in $\mathbf{x}_t$ when conditioned on $y$, and $w$ is the guidance scale, which controls how strongly the conditional information influences the generation.

By adjusting $w$, one can control the balance between sample diversity and adherence to the conditioning $y$. When $w = 1$, the process is equivalent to standard conditional generation. When $w > 1$, the conditional prediction is amplified, guiding the model to produce samples that more closely match the conditioning information, potentially at the cost of diversity.

## 4 AUTOLORA

Karras et al. (2024) introduced a novel technique called AutoGuidance to enhance the image generation capabilities of a diffusion model by guiding it with *a bad* version of itself – a smaller and less-trained variant. This method leads to more refined results while maintaining diversity in the outputs. The AutoGuidance approach is not only effective for both conditional and unconditional models but also achieves superior results without additional external models or resources for guidance. It is defined by modifying CFG in Equation 4:

$$\hat{\boldsymbol{\epsilon}}_\theta(\mathbf{x}_t, y) = \boldsymbol{\epsilon}_{\text{bad}}(\mathbf{x}_t, y) + w\left(\boldsymbol{\epsilon}_\theta(\mathbf{x}_t, y) - \boldsymbol{\epsilon}_{\text{bad}}(\mathbf{x}_t, y)\right), \tag{5}$$

where $\boldsymbol{\epsilon}_{\text{bad}}$ is an undertrained version of the base model $\boldsymbol{\epsilon}_\theta$.

The core intuition behind AutoGuidance is that the "bad" version of a diffusion model can effectively explore additional directions of the trajectories in the reverse process leading to more diverse samples. Meanwhile, the base model drives the inference towards the most probable paths and is responsible for quality of the outcomes and class-matching in case of additional conditioning. The parameter $w$ controls the exploration-exploitation trade-off of the process.

In this work, we present AutoLoRA, the method that increases the diversity and quality of generated samples of the models fine-tuned with the LoRA-based method. The core idea of this approach is to provide a sampling procedure with the trade-off between exploring the directions with the base

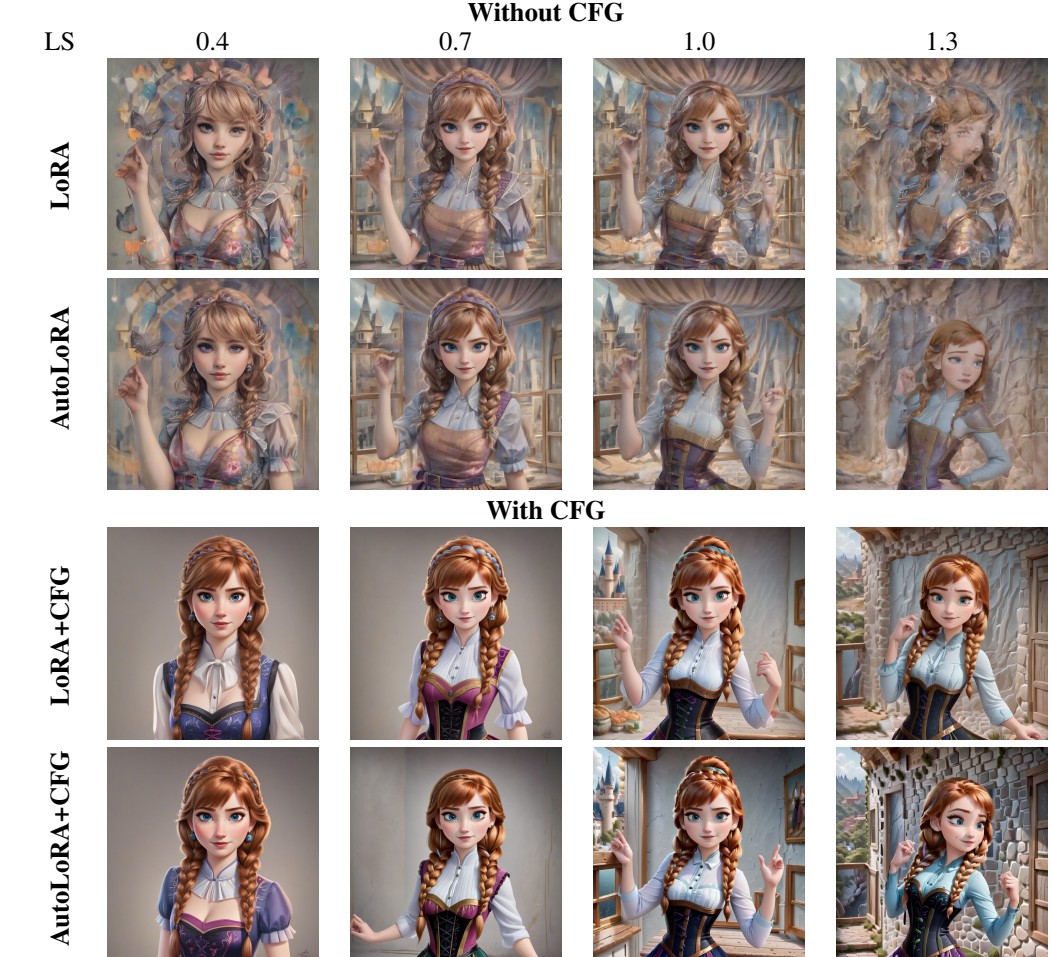

Figure 1: Comparison of the influence of different LoRA scales. Columns correspond to the scales: 0.4, 0.7, 1.0 and 1.3. All samples are generated from the same initial noise.

conditional diffusion model and being consistent with the path determined by the fine-tuned version. This type of guidance can be seen as a sort of regularisation to prevent collapsed samples generated from the model prone to overfitting during the fine-tuning stage. Moreover, to increase the quality of generated samples and diversity of the image regions not specified in the prompt separate classifier-free guidances are used for base and fine-tuned models, respectively.

Let's denote the base conditional model by $\epsilon(\mathbf{x}_t, y)$ and the same model finetuned with additional LoRA parameters as $\epsilon_{\mathrm{LoRA}}(\mathbf{x}_t, y)$. We define the AutoLoRA guidance as:

$$\epsilon^{\gamma}_{\mathrm{AutoLoRA}}(\mathbf{x}_t, y) = \epsilon(\mathbf{x}_t, y) + \gamma \cdot (\epsilon_{\mathrm{LoRA}}(\mathbf{x}_t, y) - \epsilon(\mathbf{x}_t, y)), \tag{6}$$

where $\gamma$ controls the balance between the diversity of generated samples and LoRA adaptation strength.

Sampling only using the context may lead to poor quality of the generated images. Practically, we postulate to use separate, classifier-free guidances for the base and fine-tuned variants of the diffusion model:

$$\hat{\epsilon}^{\gamma, w_1, w_2}_{\mathrm{AutoLoRA}}(\mathbf{x}_t, y) = \hat{\epsilon}^{w_1}(\mathbf{x}_t, y) + \gamma \cdot (\hat{\epsilon}^{w_2}_{\mathrm{LoRA}}(\mathbf{x}_t, y) - \hat{\epsilon}^{w_1}(\mathbf{x}_t, y)), \tag{7}$$

where $\hat{\epsilon}^{w_1}(\mathbf{x}_t, y)$ and $\hat{\epsilon}^{w_2}_{\mathrm{LoRA}}(\mathbf{x}_t, y)$ are results of classifier-free guidance given by Equation 4 with balancing parameters $w_1$ and $w_2$ respectively.

The process of reverse sampling is described by Algorithm 2. First, the classifier-free guidance is applied both for base $\epsilon(\mathbf{x}_t, y)$ and fine-tuned $\epsilon_{\mathrm{LoRA}}(\mathbf{x}_t, y)$ model variants in order to get the guided

Table 1: Comparison of diversity (from 0 to 1), VLM based Character Presence Score (CPS) (from 0 to 5) and their product Div-CPS over 512 images generated using "Anna" prompt and SDXL "Disney princesses" LoRA.

| | | | | | | | | | | | | |
|---|---|---|---|---|---|---|---|---|---|---|---|---|
| | **Without CFG** | | | | | | **With CFG** | | | | | |
| | **LoRA** | | | **AutoLoRA** | | | **LoRA+CFG** | | | **AutoLoRA+CFG** | | |
| LoRA Scale | Diversity | CPS | Div-CPS | Diversity | CPS | Div-CPS | Diversity | CPS | Div-CPS | Diversity | CPS | Div-CPS |
| 0.2 | 0.378 | 0.010 | 0.004 | 0.369 | 0.014 | **0.005** | 0.262 | 0.742 | 0.195 | 0.254 | 1.072 | **0.273** |
| 0.3 | 0.360 | 0.016 | 0.006 | 0.346 | 0.029 | **0.010** | 0.236 | 1.863 | 0.439 | 0.226 | 2.906 | **0.656** |
| 0.4 | 0.346 | 0.037 | 0.013 | 0.333 | 0.059 | **0.020** | 0.218 | 3.326 | 0.726 | 0.209 | 3.994 | **0.837** |
| 0.5 | 0.340 | 0.100 | 0.034 | 0.324 | 0.314 | **0.102** | 0.209 | 4.217 | 0.880 | 0.203 | 4.430 | **0.898** |
| 0.6 | 0.337 | 0.152 | 0.051 | 0.326 | 0.771 | **0.251** | 0.208 | 4.467 | 0.931 | 0.207 | 4.723 | **0.978** |
| 0.7 | 0.337 | 0.389 | 0.131 | 0.329 | 1.260 | **0.414** | 0.214 | 4.666 | 1.001 | 0.214 | 4.793 | **1.028** |
| 0.8 | 0.341 | 0.496 | 0.169 | 0.334 | 1.850 | **0.618** | 0.221 | 4.729 | 1.047 | 0.226 | 4.711 | **1.064** |
| 0.9 | 0.347 | 0.543 | 0.188 | 0.338 | 2.115 | **0.714** | 0.231 | 4.674 | 1.081 | 0.235 | 4.703 | **1.104** |
| 1.0 | 0.357 | 0.693 | 0.247 | 0.345 | 1.998 | **0.689** | 0.243 | 4.543 | 1.103 | 0.248 | 4.631 | **1.148** |
| 1.1 | 0.369 | 0.650 | 0.240 | 0.355 | 1.803 | **0.641** | 0.255 | 4.535 | 1.155 | 0.256 | 4.529 | **1.159** |
| 1.2 | 0.383 | 0.582 | 0.223 | 0.367 | 1.492 | **0.548** | 0.265 | 4.311 | **1.143** | 0.265 | 4.283 | 1.134 |
| 1.3 | 0.401 | 0.463 | 0.185 | 0.381 | 1.115 | **0.425** | 0.283 | 3.975 | **1.124** | 0.282 | 3.914 | 1.102 |

versions $\hat{\epsilon}^{w_1}(\mathbf{x}_t, y)$ and $\hat{\epsilon}^{w_2}_{\text{LoRA}}(\mathbf{x}_t, y)$. Next, they are used to create the final model combination with Equation 7. The final model $\hat{\epsilon}^{\gamma, w_1, w_2}_{\text{AutoLoRA}}(\mathbf{x}_t, y)$ is further used to generate sample $\mathbf{x}_{t-1}$ in the same manner as in classifier-free guidance. The procedure is repeated, starting from Gaussian noise sample $\mathbf{x}_T$ in $t = T$ that is iteratively transformed to the sample $\mathbf{x}_0$ from data distribution.

## 5 EXPERIMENTS

To present the capabilities and effectiveness of AutoLoRA in guiding large diffusion models towards the direction of a subspace of desired images, we perform a wide set of experiments comparing our methods against standard approaches. In subsequent sections, we initially discuss the metrics used to measure quality and diversity, incorporating established and new metrics. Next, we outline the general configuration of all conducted experiments. Finally, we delve into the experimental details and the results obtained.

### 5.1 EVALUATION METRICS

To quantitatively evaluate the diversity and consistency of our methods we postulate to utilize the cosine similarity-based metric. We quantify **diversity** as follows:

$$\text{Diversity}(\mathbf{X}) = 1 - \frac{2}{N(N-1)} \sum_{i=1}^{N} \sum_{j=i+1}^{N} \text{cosine\_similarity}(f_\theta(\mathbf{x}_i), f_\theta(\mathbf{x}_j)), \tag{8}$$

where $\mathbf{X}$ is a set of $N$ samples generated using the same prompt and $f_\theta$ is an image feature extractor. As you can notice the diversity metric reaches maximum value if the samples are totally different. However, in addition to the samples' variety the correspondence to the prompt is also important so we utilize the Visual Language Models to measure:

**Character Presence Score (CPS)** is a quantitative measure that evaluates the presence of a specific character in an image with scores ranging from 0 (not present) to 5 (unmistakably present).

**Prompt Correspondence (PC)** measures how well an image captures the essence, objects, and scenes described in the original prompt with scores ranging from 0 (not at all) to 5 (exactly).

**Style Adherence (SA)** assesses how well an image adheres to a specified style with scores ranging from 0 (markedly deficient) to 5 (perfectly).

To further evaluate the generated images, we introduce novel metrics that combine the diversity metric with the above measures, namely **Div-CPS**, **Div-PC**, and **Div-SA**, which are calculated by multiplying the diversity metric with the CPS, PC, and SA, respectively.

## 5.2 EXPERIMENTAL SETUP

In all experiments, we are using the current state-of-the-art large diffusion models - Stable Diffusion XL (SDXL)(Podell et al., 2023) and Stable Diffusion 3 Medium (SD3) (Esser et al., 2024), which are open-source and highly available, e.g., via HuggingFace[1].

**LoRA modules** For the fair comparison of AutoLoRA and baseline methods, we selected a set of a few diverse LoRA modules, which were highly rated on the known community site civitai[2]. In specific, for the detailed diversity evaluation of different diffusion model parameters, we choose *"All Disney Princess XL LoRA Model from Ralph Breaks the Internet"*[3] with SDXL backbone. Whereas, for the less extensive evaluation, we compare SDXL with *"Pixel Art XL - v1.1"*[4] and SD3 with *"Pixel Art Medium 128 v0.1"*[5].

For all Visual Language Model-based evaluations we use llama-3.2-11B-Vision-Instruct model (Dubey et al., 2024)[6]. By default, the scale parameter for Classifier-Free Guidance (CFG) is set to 5.0 for SDXL, 7.0 for SD3, and scale factor for AutoLoRA is 1.5. Finally, for the feature extraction in the diversity quantification, we utilize DINOv2 (Oquab et al., 2023)[7] model.

## 5.3 DISNEY PRINCESS

In this subsection, we gradually show the combinations of AutoLoRA with different diffusion model parameters over **Disney Princess** LoRA for SDXL. Firstly, we explore the influence of the LoRA scale parameter. Secondly, we investigate the effects of different CFG scales and finally, we demonstrate how AutoLoRA scale can improve the generated images.

For the detailed analysis of AutoLoRA's performance in combination with different diffusion model parameters such as LoRA scale, and Classifier-Free Guidance (CFG) scale, we chose **Disney Princess** LoRA for SDXL and generated images using the prompt *"Anna"* that corresponds to the character of Princess Anna from the movie Frozen. This prompt is generic enough that the base SDXL model does not produce the desired character and allows the LoRA model to generate *"Anna"* in different settings so that we can observe the variety of outputs.

In Table 1, we present the effect of the LoRA scale in 4 scenarios: vanilla LoRA. LoRA guided by AutoLoRA without CFG, LoRA model with CFG, and AutoLoRA combined with CFG. As you can notice, without CFG the model with tuned LoRA weights generates highly diverse outputs; however, they often do

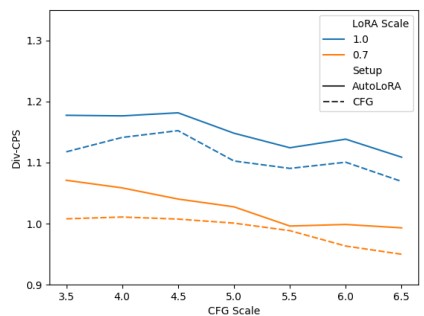

Figure 2: Comparison of the impact of different Classifier-Free Guidance scales for the vanilla LoRA model with CFG and AutoLoRA.

not include the desired character as also shown in Figure 1. For the CFG case, the increase in the LoRA scale makes the model leverage the Disney style more and produce a more recognizable Anna. Additionally, samples generated by AutoLoRA guidance demonstrate much more details and beat the basic CFG results.

For further comparison between CFG and AutoLoRA techniques, we investigated the influence of the CFG scale change. As depicted in Figure 2, AutoLoRA consistently overperforms the vanilla CFG approach. The qualitative comparison is presented in Figure 3.

---

[1]https://huggingface.co

[2]https://civitai.com

[3]https://civitai.com/models/212532/all-disney-princess-xl-lora-model-from-ralph-breaks-the-internet

[4]https://civitai.com/models/120096/pixel-art-xl?modelVersionId=135931

[5]https://huggingface.co/nerijs/pixel-art-medium-128-v0.1

[6]https://huggingface.co/meta-llama/Llama-3.2-11B-Vision-Instruct

[7]https://huggingface.co/facebook/dinov2-giant

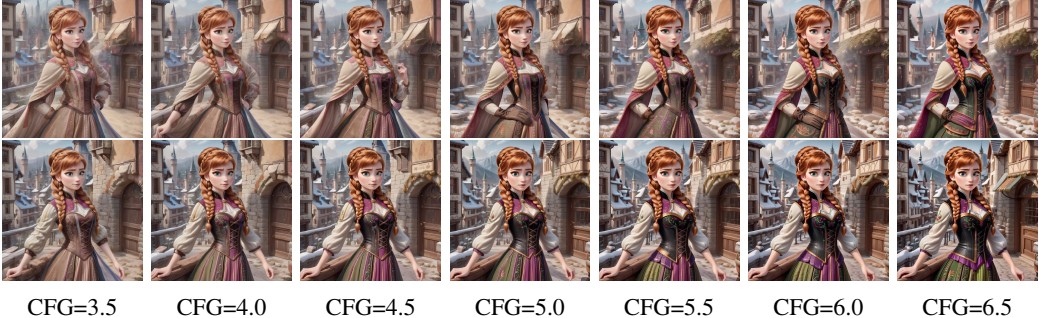

| CFG=3.5 | CFG=4.0 | CFG=4.5 | CFG=5.0 | CFG=5.5 | CFG=6.0 | CFG=6.5 |

Figure 3: Comparison of the influence of different Classifier-Free Guidance scales for the vanilla LoRA model with CFG (*top*) and AutoLoRA (*bottom*). Below the samples, we presented the corresponding value of a CFG scale parameter. All samples are generated from the same initial noise.

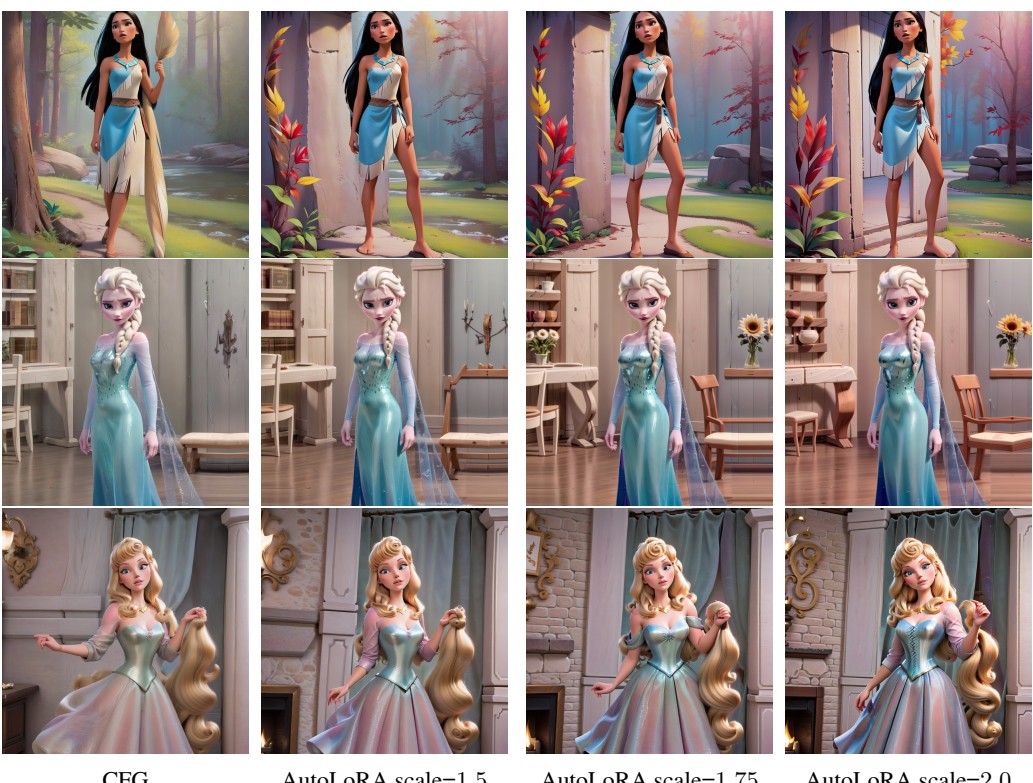

| CFG | AutoLoRA scale=1.5 | AutoLoRA scale=1.75 | AutoLoRA scale=2.0 |

Figure 4: Comparison of different scale factors (*i.e.,* 1.5, 1.75, 2.0) in AutoLoRA. Samples in each row are generated from the same noise initial noise using Stable Diffusion XL and Disney Princess LoRA. CFG samples are presented on the *left* column as a reference.

Figure 4 shows the dependence between AutoLoRA scale and the diversity of the generated samples. The higher the used scale, the more details appear on the images.

## 5.4 PIXEL-ART

In this subsection, we present the quantitative and qualitative comparison of the vanilla LoRA model with CFG and AutoLoRA using SDXL with **Pixel Art XL v1.1** LoRA and SD3 with **Pixel Art Medium 128 v0.1** LoRA.

**Stable Diffusion XL**  **Stable Diffusion 3 Medium**
CFG           AutoLoRA           CFG           AutoLoRA

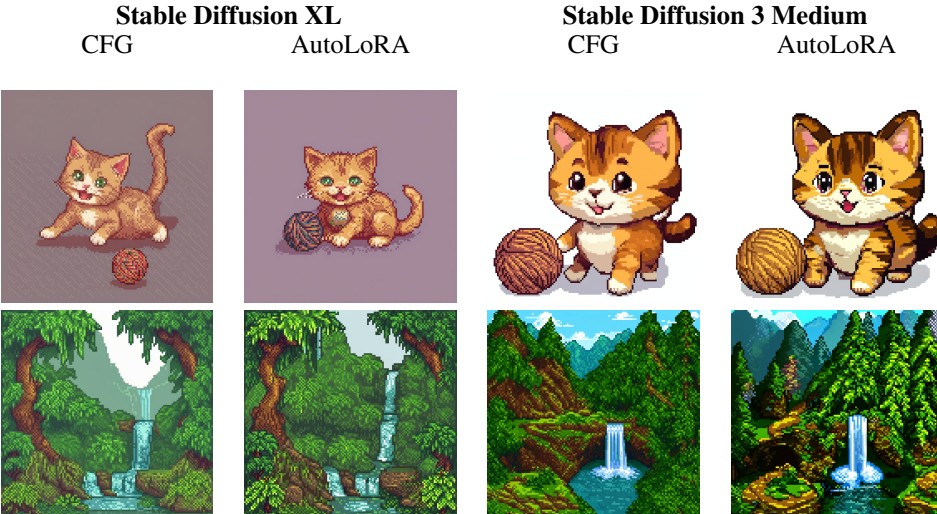

Figure 5: Samples from the same noise using CFG and AutoLoRA for Pixel Art LoRA module and two models – Stable Diffusion XL (*left* columns) and Stable Comparison 3 (*right* columns).

Table 2: Comparison of diversity (from 0 to 1), VLM-based Prompt Correspondence (PC) and Style Adherence (SA) metrics, and their corresponding products Div-PC and Div-SA over 480 images (30 different prompts and 16 images per prompt)

| Method | Diversity | Prompt Correspondence | Style Adherence | Div-PC | Div-SA |
|---|---|---|---|---|---|
| SD3 CFG | 0.136 | **3.985** | **4.177** | 0.540 | 0.566 |
| SD3 AutoLoRA | **0.197** | 3.725 | 4.019 | **0.734** | **0.792** |
| SDXL CFG | 0.159 | **3.846** | 4.144 | 0.611 | 0.658 |
| SDXL AutoLoRA | **0.170** | 3.756 | **4.150** | **0.637** | **0.704** |

We used a set of 30 diverse prompts and generated 16 images per prompt. We leverage the same set of random seeds for all 4 evaluated configurations: SDXL CFG, SDXL AutoLoRA, SD3 CFG, and SD3 AutoLoRA. As indicated in Table 2, AutoLoRA beats the CFG in terms of diversity and normalized diversity metrics as Div-PC and Div-SA. In Figure 5, we demonstrate how AutoLoRA produces more details in comparison to the CFG technique.

## 6 CONCLUSION

This paper presents AutoLoRA, an innovative guidance method for diffusion models tailored with the LoRA approach. Drawing inspiration from other guidance techniques, AutoLoRA aims to balance consistency within the domain highlighted by LoRA weights and the variety of samples from the primary conditional diffusion model. Additionally, we demonstrate that using distinct classifier-free guidances for the LoRA fine-tuned and base models enhances the diversity and quality of generated samples. Experimental results in various fine-tuned LoRA domains indicate that our method outperforms current guidance techniques across selected metrics.

**Limitation and Future Work**  The naive implementation AutoLoRA used in the paper increases the inference time ×2 because for each diffusion step, we run LoRA tuned and base models. However, this issue could be solved using model distillation techniques. In future works, we also plan to extend our approach with the idea of applying AutoLoRA in a limited interval.

**Reproducibility Statement**  We will release the code in a camera-ready version. All images were generated using the fixed set of seeds. For the VLM-based evaluations, you can find the exact prompts in the Appendix section A. All prompts used for the pixel art style images are included in Appendix section B.

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

In this appendix, we start by providing the Visual Language Model prompts used in the experiments in Section A. Then, we include all prompts used for the pixel art style image generation in Section B. Next, we demonstrate additional qualitative results to show the superiority of AutoLoRA approach in Section C.

# A    VISUAL LANGUAGE MODEL EVALUATION PROMPTS

## A.1    ANNA PRESENCE SCORE

<IMAGE DATA> Evaluate the presence of Princess Anna from Disney's Frozen movie in the image. Output a score between 0 and 5, where:
* 0: Princess Anna is not present in the image.
* 1: The image contains a character with a vague resemblance to Princess Anna, but it's not clear if it's her. (e.g., a character with a similar hairstyle or dress color)
* 2: The image contains a character that shares some similarities with Princess Anna, but it's not a clear match. (e.g., a character with a similar face shape or clothing style)
* 3: The image contains a character that is similar to Princess Anna, but with some noticeable differences. (e.g., a character with a similar dress and hairstyle, but different facial features)
* 4: The image contains a character that is very likely to be Princess Anna, but with some minor differences. (e.g., a character with a similar face, dress, and hairstyle, but with a slightly different expression or pose)
* 5: The image contains a character that is unmistakably Princess Anna from Frozen.

Output the score in following JSON format:
{
"score": [score between 0 and 5],
"reason": [use keywords to describe the reason of the score, e.g., ["dress", "hairstyle", "no Anna character"] ]
}
Reply only with a JSON with no extra text

## A.2    PIXEL-ART PROMPT CORRESPONDENCE AND STYLE ADHERENCE

<IMAGE DATA> Evaluate the given image for the prompt <PROMPT USED FOR THE IMAGE GENERATION> and the pixel art style
Access the following metrics:
Prompt correspondence: How well does the image capture the essence, objects, and scenes described in the prompt? Scale: 0-5, where:
0: Not at all (the image does not relate to the prompt in any way)
1: Very poorly (the image vaguely relates to the prompt, but most key elements are missing or incorrect)
2: Somewhat (the image captures some key elements of the prompt, but others are missing or incorrect)
3: Fairly well (the image captures most key elements of the prompt, but some details may be off)
4: Very well (the image accurately captures the essence and most key elements of the prompt)
5: Exactly (the image perfectly captures the essence, objects, and scenes described in the prompt)
Style adherence: How well does the image adhere to the specified style? If the style is pixel art, does the image truly resemble pixel art, or is it just a low-quality image? Scale: 0-5, where:
1: Very poorly (the image attempts to mimic pixel art, but lacks clear pixelation, has excessive aliasing, or uses too many colors)
2: Somewhat (the image shows some pixel art characteristics, such as pixelation, but lacks consistency in pixel size, color palette, or has noticeable artifacts)
3: Fairly well (the image generally adheres to pixel art principles, with clear pixelation, a limited color palette, and minimal aliasing, but may have some minor flaws)
4: Very well (the image strongly adheres to pixel art principles, with crisp pixelation, a well-chosen color palette, and minimal to no aliasing or artifacts)
5: Perfectly (the image perfectly captures the pixel art style, with precise pixelation, a masterfully chosen color palette, and no noticeable flaws or artifacts)
Provide the evaluation scores in the following JSON format:

```
{
"prompt_correspondence": [the prompt correspondence score from 0 to 5],
"style_adherence": [the style adherence score from 0 to 5],
}
```
Reply only with a JSON with no extra text

## B  PIXEL ART PROMPTS

a dragon riding a unicorn through a rainbow-colored sky, pixel art style
a mermaid sitting on a throne made of coral and seashells, pixel art style
a phoenix rising from a pile of ashes, surrounded by flames, pixel art style
a centaur archer aiming at a target in a mystical forest, pixel art style
a gryphon guarding a treasure chest filled with glittering jewels, pixel art style
a futuristic city on a distant planet, with towering skyscrapers and flying cars, pixel art style
a robot astronaut exploring a desolate, alien landscape, pixel art style
a spaceship battling a giant, tentacled space monster, pixel art style
a cyborg warrior standing on a barren, post-apocalyptic wasteland, pixel art style
a group of aliens enjoying a picnic on a sunny, grassy hill, pixel art style
a giant, juicy burger with all the toppings, surrounded by condiments and fries, pixel art style
a steaming hot cup of coffee, with a dash of cream and a sprinkle of cinnamon, pixel art style
a colorful, layered cake with candles and festive decorations, pixel art style
a plate of sushi, with a variety of rolls and sashimi, pixel art style
a glass of sparkling champagne, with a champagne flute and strawberries, pixel art style
a proud lion standing on a rocky outcropping, with a savannah landscape behind, pixel art style
a school of rainbow-colored fish swimming together in unison, pixel art style
a majestic, snow-white owl perched on a branch, surrounded by moonlight, pixel art style
a happy, playful kitten chasing a ball of yarn, pixel art style
a serene, peaceful landscape with a waterfall and lush greenery, pixel art style
a classic, old-fashioned video game arcade, with pixel art cabinets and joysticks, pixel art style
a vintage, convertible car driving down a sunny, palm-lined road, pixel art style
a retro-style, neon-lit diner, with a jukebox and milkshakes, pixel art style
a cassette tape player, with a mix tape and a pair of headphones, pixel art style
a payphone, with a rotary dial and a busy city street behind, pixel art style
a swirling, psychedelic vortex, with colors and patterns blending together, pixel art style
a dreamlike, surreal landscape, with melting clocks and distorted objects, pixel art style
a geometric, fractal pattern, with repeating shapes and colors, pixel art style
a kaleidoscope of colors, with shifting, mirrored reflections, pixel art style
a stylized, abstract representation of a musical waveform, with vibrant colors and shapes, pixel art style

## C ADDITIONAL QUALITATIVE RESULTS

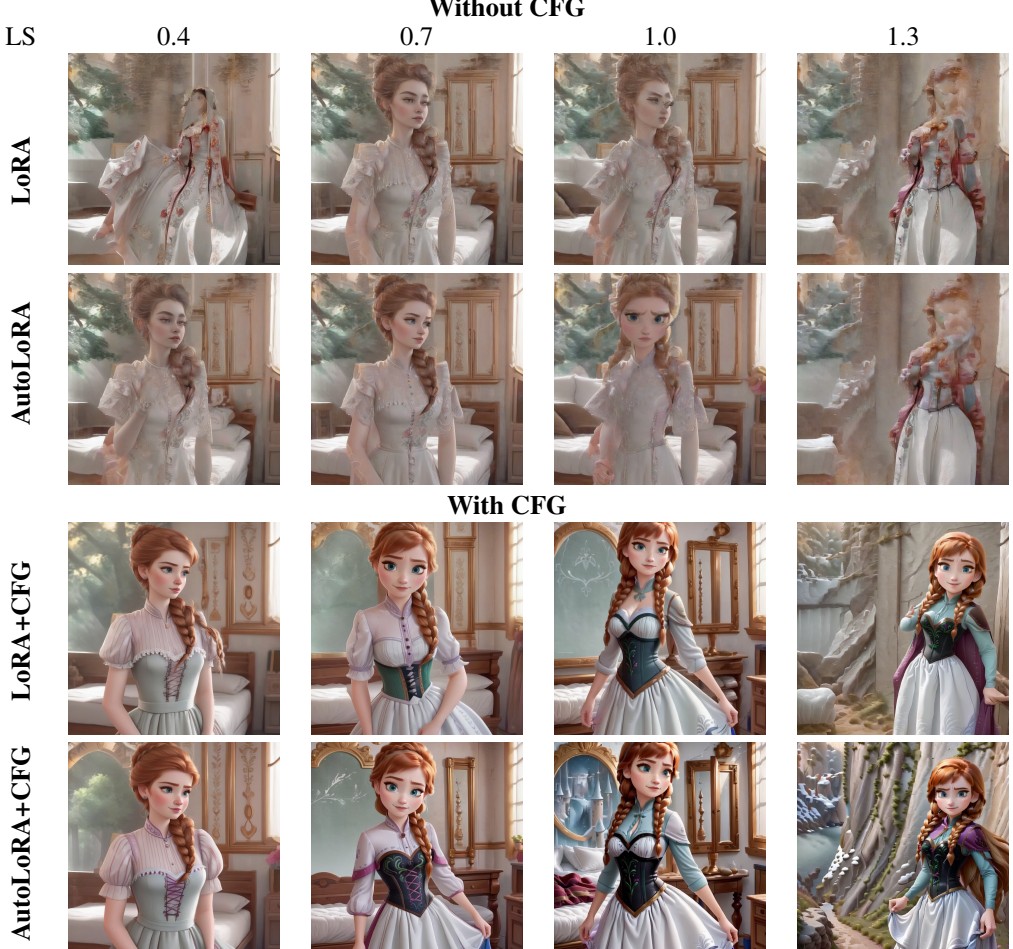

Figure 6: Comparison of the influence of different LoRA scales. Columns correspond to the scales: 0.4, 0.7, 1.0 and 1.3. All samples are generated from the same initial noise.

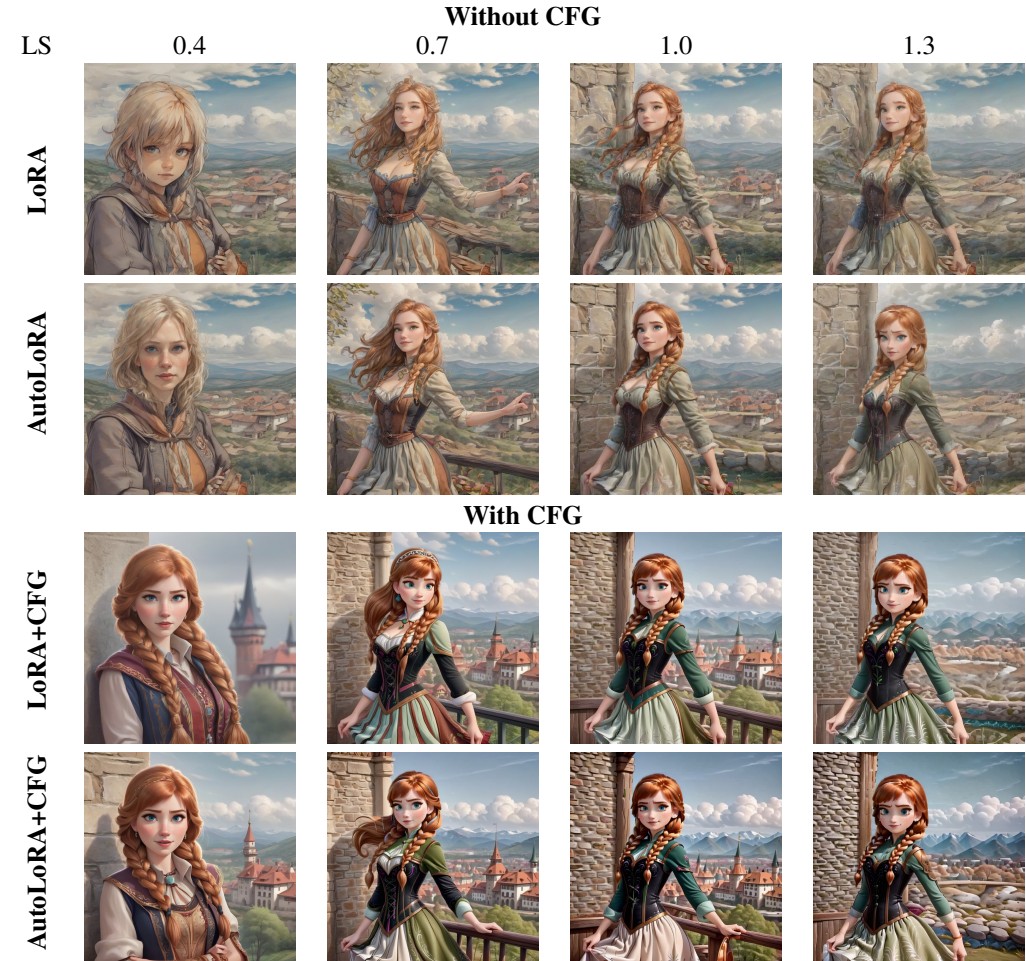

Figure 7: Comparison of the influence of different LoRA scales. Columns correspond to the scales: 0.4, 0.7, 1.0 and 1.3. All samples are generated from the same initial noise.

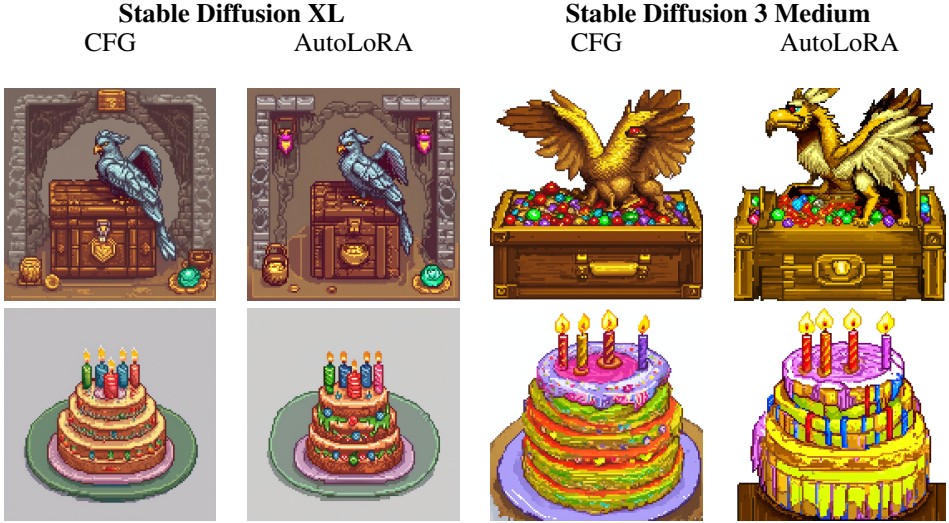

Figure 8: Samples from the same noise using CFG and AutoLoRA for Pixel Art LoRA module and two models – Stable Diffusion XL (*left* columns) and Stable Comparison 3 (*right* columns)

