# OpenReview forum: "AutoLoRA: AutoGuidance Meets Low-Rank Adaptation for Diffusion Models"
_ICLR.cc/2025/Conference — ICLR 2025 Conference Withdrawn Submission_

### Official Review · Reviewer_8otW · 2024-10-30

**Soundness:** 2
**Presentation:** 1
**Contribution:** 1
**Rating:** 3
**Confidence:** 3

**Summary:**

While LoRA-based fine-tuning for diffusion models can achieve comparable results to full fine-tuning with significantly few parameters. The generated images usually shows strong context bias and low diversity. To address this issue, this paper proposes AutoLoRA, which leverage the idea of AutoGuidance to increase diversity and image quality. Specifically, they use the base model before LoRA tuning to condition the final variant  fine-tuned with the LoRA approach. Experimental results demonstrate that AutoLoRA outperforms existing guidance techniques across multiple domains, showcasing superior diversity and alignment.

**Strengths:**

The proposed method is simple, easy to implement, and practical for the community.

**Weaknesses:**

(1) The motivation is unconvincing. The authors state in Lines 44–45 that LoRA reduces the diversity of generated images. However, Table 1 shows that LoRA demonstrates higher diversity across different LoRA scales compared to the proposed AutoLoRA. I recommend that the authors revise the introduction to ensure consistency in their arguments. The proposed method appears trivial, as it is merely a straightforward combination of LoRA and AutoGuidance. I recommend that the authors provide further justification for why combining these two techniques is particularly suitable for fine-tuning diffusion models.


(2) The qualitative results are not visually superior. From Figures 1, 3, and 4, it is unclear which method performs better visually. I suggest that the authors provide additional explanations in the captions to clarify the comparisons.


(3) Lack of Ablation studies for w_1 and w_2. The authors use separate classifier-free guidance for the based and the fine-tuned variants. However, the ablation studies about how the two hyper-parameters influence the results are missing. I suggest the authors to include them in the manuscript.

(4) The writing and structure could be improved. For example, the related work section is too lengthy, spanning approximately one and a half pages, and should be more concise.

**Questions:**

Please refer to the weaknesses.

---

### Official Review · Reviewer_fCiA · 2024-11-03

**Soundness:** 2
**Presentation:** 2
**Contribution:** 3
**Rating:** 3
**Confidence:** 3

**Summary:**

This paper presents AutoLoRA, a novel technique that combines AutoGuidance with LoRA for diffusion models. The primary goal of AutoLoRA is to improve the sample diversity of diffusion models fine-tuned with LoRA by introducing a guidance method that balances LoRA-specific consistency with broader sample diversity. This is achieved through a dual-layer guidance strategy where both LoRA-tuned and base diffusion models apply classifier-free guidance (CFG). The main contribution of this work is a new AutoLoRA guidance mechanism that adapts LoRA-fine-tuned models to improve diversity and quality in generated samples.

**Strengths:**

1.	Originality: The paper introduces a unique combination of AutoGuidance and LoRA for diffusion models, expanding on traditional guidance approaches. The new inference scheme can achieve a higher generated image diversity while maintaining prompt correspondence.
2.	Quality: The study is methodologically sound, detailing the AutoLoRA algorithm and presenting clear mathematical formulations. The experiments include comparisons across different diffusion models and LoRA modules, and they explore how AutoLoRA parameters influence diversity, which is essential for understanding model behavior.
3.	Clarity: The paper provides a comprehensive explanation of AutoLoRA, with well-organized sections on preliminary, methodology, evaluation metrics, and experimental outcomes.
4.	Significance: AutoLoRA represents an advancement for conditional image generation, offering a way to mitigate the diversity problem in LoRA fine-tuned models. This would be important for the LoRAs that highly overfit to specific domains.

**Weaknesses:**

1.	There are some passages affecting the fluency of reading the paper. For example, the AutoGuidance in the Introduction shall be discussed after introducing the problem the authors want to solve instead of at the very beginning. The related work and preliminaries are too detailed, decreasing the importance of the proposed method.
2.	The quantitative improvements provided by AutoLoRA over CFG (e.g., in Div-CPS and Div-SA) are not substantial across all experimental setups. The experiments are centered primarily on specific models (e.g., Stable Diffusion) and particular LoRA modules like the "Disney Princess" and "Pixel Art" LoRA. More detailed results across diverse LoRAs and prompts would strengthen claims of AutoLoRA’s effectiveness. In addition, the improvements in SDXL (Table 2) seem not to be surprising, considering the 2x inference time compared to the baseline.
3.	The dual-model requirement, where both the base and LoRA models are applied at each diffusion step, increases the inference complexity significantly. With nested guidance, the model’s performance shall be compared to that of more advanced methods instead of only the baseline.
4.	The paper introduces new metrics (Div-CPS, Div-SA) without clear validation or benchmarks to assess their reliability as indicators of model performance. By multiplying the diversity metric with CPS and SA, their magnitude themselves would affect the final result to a different scale. For example, the diversity ranges from 1.0 to 2.0, but SA has a much smaller range from 4.0 to 4.2. Would you consider the normalization of the two quantities a better way?
5.	Some visualizations cannot support the claims. For example in Figures 3 and 4, readers can hardly tell the performance difference between CFG and AutoLoRA.

**Questions:**

1.	Could the authors elaborate on how the AutoLoRA parameter $\gamma$ impacts the diversity-quality tradeoff in different settings? It would be helpful to understand if there is an optimal range of $\gamma$ for specific scenarios.
2.	Since the current setup increases inference time, do the authors plan to compare the performance between AutoLoRA and other more sophisticated methods? A simple but fair comparison is sampling 2x images to increase the diversity for baseline.
3.	Can the authors provide comparisons with other metrics (e.g., ImageReward, HPS) or benchmarks that validate their effectiveness in capturing diversity and prompt adherence?
4.	Given the experimental results, how does AutoLoRA handle prompts with highly complex or abstract styles? Extending the experiments to such prompts could provide insights into the model's capability to generalize beyond straightforward character or style constraints.

---

### Official Review · Reviewer_cZJq · 2024-11-03

**Soundness:** 3
**Presentation:** 3
**Contribution:** 2
**Rating:** 5
**Confidence:** 4

**Summary:**

This paper proposes AutoLoRA,  a novel guidance technique that combines LoRA fine-tuning with an autoguidance strategy in diffusion models to generate more diverse samples. By incorporating classifier-free guidance, AutoLoRA produces samples with higher diversity and better quality.

**Strengths:**

1. The paper is well written and easy to follow.
2. This paper presents a novel guidance technique on finetune-based diffusion models, which combines LoRA with  an autoguidance strategy to improve the quality and diversity of generated samples.
3. This paper propose some reasonable metrics to evaluate the diversity and consistency of generated samples, including Diversity, Character Presence Score (CPS), Prompt Corresepondence (PC) and Style Adherence (SA).

**Weaknesses:**

1. It appears that the paper lacks a theoretical analysis explaining the design of AutoLoRA and why it is effective in balancing the diversity of the original pre-trained model with the consistency with LoRA-tuned new model. Providing a more in-depth theoretical justification for AutoLoRA would help to differentiate it with Autoguidance.

2. It looks that the paper also lacks some comaprasion experiments with other fine-tuning diffusion models, such as DreamBooth, Textural Inversion. Furthermore, the paper lacks general metrics for evaluating image-alignment and text-alignment. Referencing metrics used in DreamBooth could provide a more standardized and comparable evaluation of AutoLoRA’s performance in these areas.

3.  Very minor points: there are some small typos. E.g., some quotation marks are not correctly used.

**Questions:**

Could you theoretically analyse the difference betweent Eq. 5 and your designed AutoLoRA?

By comparing these two equations, it seems to me that the main difference between Eq. 5 and AutoLoRA is the choice of the "bad" and "better" version models. In Eq. 5, the "bad" version model is a smaller and less-trained diffusion model, while in AutoLoRA, the "bad" version model is the pre-trained diffusion model and the LoRA-tuned model as the better version model. However, the lack of diversity in the LoRA-tuned model may limit the overall diversity of the generated samples. Thus, using LoRA-tuned model as the "better" version model may not be a reasonable choice to me.

---

### Official Review · Reviewer_Qi7S · 2024-11-04

**Soundness:** 1
**Presentation:** 1
**Contribution:** 1
**Rating:** 1
**Confidence:** 4

**Summary:**

This paper proposes a refinement of diffusion models finetuned with LoRA. Starting with the observation that the finetuned model has limited diversity in the generated images due to limited data, the paper proposes a trade-off between the base and LoRA finetuned model by conditioning LoRA with the base model.

**Strengths:**

The core idea is to provide a sampling procedure with the trade-off between exploring the directions with the base conditional diffusion model consistent with the path determined by the LoRA finetuned version, similar to classifier-free guidance.

**Weaknesses:**

I carefully reviewed the paper, and rather than addressing an extensive list of less critical and debatable issues, I will concentrate on the most significant concerns that influence my rating. My focus will be on the following three points:

1) The novelty is very marginal. The idea is a straightforward extension of AutoGuidance, yet instead of conditional and unconditional, here, the guidance is based on the base and LoRA finetuned models.

2) The diversity improvement is questionable and insignificant. The only metric that indicates some improvement, DiV-CPS, evaluates the presence of a specific character in an image. It is not clear how the CPS score is computed and whether it is accurate or not. How are such specific characters defined? How stylizing LoRA's use character presence? This metric seems completely unworkable. Besides the visual comparisons provided (e.g., Figure 1) do not make a case for any diversity improvement with AutoLoRA.

3) Language is poor, and the discussion does not provide insights into parameter choices. Most of the presentation is dedicated to preliminary and well-known models.

I am open to reconsidering my rating based on a satisfactory rebuttal from the authors.

**Questions:**

See the weaknesses section.

**Details Of Ethics Concerns:**

LoRA finetuning has potential to create privacy and safety concerns. Paper does not provide an ethics statement.

---

### Note · Authors · 2024-11-20

**Comment:**

Thank you for your time and effort in reviewing. We have decided to withdraw the paper to incorporate the suggested changes.

**Withdrawal Confirmation:**

I have read and agree with the venue's withdrawal policy on behalf of myself and my co-authors.